MINIREVIEW
# Cell-Cell Signaling Proteobacterial LuxR Solos: a Treasure Trove of Subgroups Having Different Origins, Ligands, and Ecological Roles

Cristina Bez,[a] Alexander Martin Geller,[b] Asaf Levy,[b] Vittorio Venturi[a]

[a]International Centre for Genetic Engineering and Biotechnology, Trieste, Italy
[b]Department of Plant Pathology and Microbiology, Institute of Environmental Science, Robert H. Smith Faculty of Agriculture, Food, and Environment, The Hebrew University of Jerusalem, Rehovot, Israel

**ABSTRACT** Many proteobacteria possess LuxR solos which are quorum sensing LuxR-type regulators that are not paired with a cognate LuxI-type synthase. LuxR solos have been implicated in intraspecies, interspecies, and interkingdom communication by sensing endogenous and exogenous acyl-homoserine lactones (AHLs) as well as non-AHL signals. LuxR solos are likely to play a major role in microbiome formation, shaping, and maintenance through many different cell-cell signaling mechanisms. This review intends to assess the different types and discuss the possible functional roles of the widespread family of LuxR solo regulators. In addition, an analysis of LuxR solo types and variability among the totality of publicly available proteobacterial genomes is presented. This highlights the importance of these proteins and will encourage scientists to mobilize and study them in order to increase our knowledge of novel cell-cell mechanisms that drive bacterial interactions in the context of complex bacterial communities.

**KEYWORDS** AHLs, LuxR solos, cell-cell interaction, microbial communities, proteobacteria, quorum sensing

Bacteria produce a variety of cell-cell contact-independent signal molecules involved in intraspecies, interspecies, and interkingdom signaling. Quorum sensing (QS) is a system of intercellular signaling that bacteria use to coordinate group behaviors, mainly among related cells (1, 2). Many proteobacteria employ QS based on acyl-homoserine lactone (AHL) signals; these share a conserved lactone core ring with various acyl chains of different lengths ($C_4$ to $C_{20}$), different oxidation states at position $C_3$, and degrees of unsaturation (3). An archetypical AHL QS system consists of two proteins: a LuxI-type AHL synthase and a LuxR-type receptor/regulator, which binds the cognate AHL and consequently regulates the expression of target genes (4).

Typically, the *luxI* and *luxR* homologs are almost always genetically linked either on the chromosome or plasmids within a certain distance (less than 3,000 bp) (3, 5). They can be arranged in different topologies; the two most common are the tandem arrangement (both genes *luxI* and *luxR* on the same strand) and the convergent arrangement ("bidirectional" genes head-to-head or tail-to-tail located on opposite strands and transcribed divergently) (6–8). In some more complex arrangements, there is one gene between the *luxR* and *luxI* genes, which frequently encodes a negative regulator, such as *rsaL* and *rsaM* genes, and in some other rare scenarios, several genes are located between the *luxR* and *luxI* genes (6). Only very few examples exist where the *luxI*-family gene is genetically unlinked from the cognate *luxR*-family gene (5). The *luxI/R* pairs likely shared a common origin coevolving as regulatory cassettes as reported by studies implementing statistical methods and covariation within their sequences (9). Many bacterial strains contain multiple *luxI/luxR* homologues acquired from an independent source via horizontal gene transfer (5, 10, 11).

Address correspondence to Vittorio Venturi, vittorio.venturi@icgeb.org.

The authors declare no conflict of interest.

It is common for bacteria to share and acquire novel regulatory genes and systems that are subsequently integrated within preexisting regulatory pathways (10). The variability of AHL-QS circuits is further enhanced by the presence of unpaired QS-related LuxR-family proteins for which there is no cognate LuxI synthase; these have been called "LuxR orphans" and, more recently, "LuxR solos" (12–14). The LuxR solos are very closely related and descended evolutionarily from the canonical QS AHL-binding LuxR-family proteins, or *vice versa*.

The number of *luxR* solo genes in a genome or metagenome is often much higher than that of *luxI*-family genes suggesting that they are very common (15, 16). Currently, few and fragmented data are available on their distribution, conservation, and ecological role among proteobacteria. The few LuxR solos studied so far highlight that they are diverse in the type of ligands they respond to as well as their target genes (14). It is therefore timely to recognize the importance of LuxR solos and their evolution away from the canonical AHL-binding regulators, allowing bacteria to respond to a wide variety of cell-cell signals and to extend bacterial cell-cell signaling and responses.

In this review we assess current knowledge on LuxR solo regulators, which we have subdivided here into 5 classes as depicted in Fig. 1; these are (A) LuxR solos responding to endogenous AHL signals, (B) LuxR solos responding to exogenous AHL signals, (C) LuxR solos responding to non-AHL endogenous signals, (D) LuxR solos responding to non-AHL exogenous signals, and (E) LuxR solos acting in a ligand-independent manner. In addition, we present an *in silico* analysis of LuxR solos in 31,034 currently publicly available proteobacterial genomes and organize their classification into different subgroups based on sequence homology and genetic features that are likely to reflect ligand specificities. A correlation of the frequency of specific LuxR solo types with the ecological niche and surrounding environment will also be discussed. Consequently, an all-inclusive picture, analysis, and review of all LuxR solos in proteobacteria is presented.

## FROM QS AHL LuxR-FAMILY PROTEINS TO LuxR SOLOS

AHL QS LuxR-family transcriptional proteins are found exclusively in proteobacteria and bind AHLs with a stoichiometric ratio of one molecule of protein to one molecule of AHL (17). LuxRs are 230 to 270 amino acids long, consisting of two modular conserved domains, an N-terminal AHL-binding domain (defined by the pfam03472), and a C-terminal DNA-binding helix-turn-helix domain (defined by the pfam00196) separated by a short linker region (18, 19). They bind DNA in a dimeric state recognizing a dyad symmetric sequence called *lux box* located in promoter regions of target genes.

They are mostly activators recruiting RNA polymerase to the promoter initiating transcription; the association with AHLs induces dimerization and results in a conformational change, enabling the activator to bind DNA (20–23). Examples of well-studied QS LuxRs acting as activators are TraR, LuxR, RhlR, and LasR (22–24). Few QS LuxRs act as transcriptional repressors in the absence of the cognate ligand, adopting the dimeric active conformation and sterically blocking transcription initiation by RNA polymerase. Examples are EsaR from *Pantoea stewartii*, ExpR from *Erwinia chrysanthemi*, and VirR from *Erwinia carotovora* (25–27). Phylogenetic analyses show that they are subdivided into two different families; proteins of family A are activators, whereas proteins of family B act as repressors (11). Surprisingly, AHL LuxR homologs display low primary sequence similarity (18% to 25%); however, the primary structures of the two domains display some highly conserved residues (5, 16). Specifically, nine amino acid residues are highly conserved; six of these are hydrophobic or aromatic and form the cavity of the AHL-binding domain (W57, Y61, D70, P71, W85, and G113 with respect to the TraR amino acid sequence). The remaining three conserved amino acids are found in the helix-turn-helix-binding domain (E178, L182, and G188) (28). QS LuxRs are very selective for their cognate AHL signal; however, there are several examples of QS LuxR-type receptors that respond promiscuously to multiple signals, having biological implications in interspecies signaling (29).

Many proteobacterial genomes possess QS-type LuxR sensors/regulators which lack a cognate LuxI AHL synthase, as already mentioned above, and these have been called LuxR

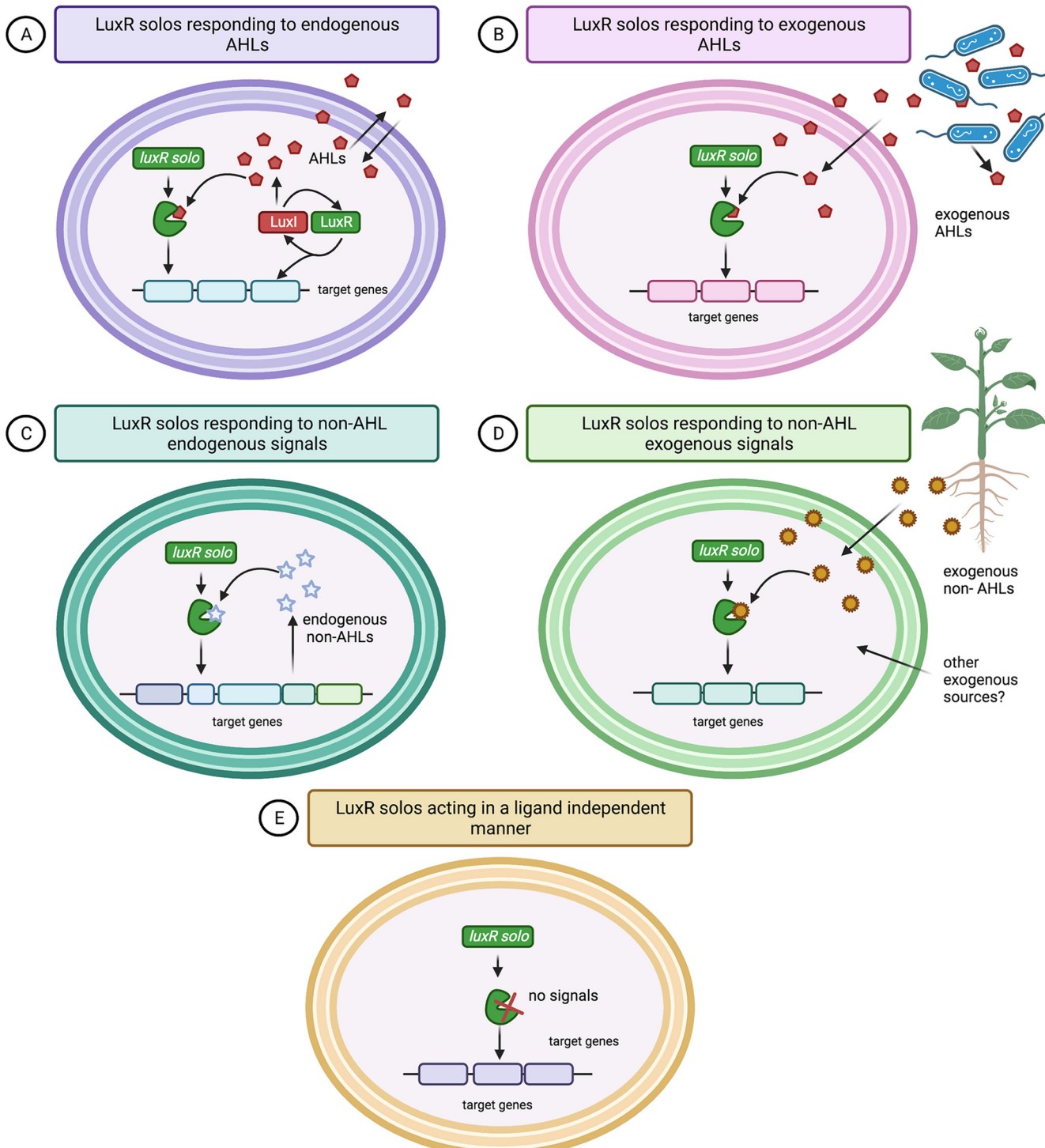

**FIG 1** Diagrammatic representation of the five functional classes in which LuxR solo types can be classified according to the nature and origin of the ligand-binding molecules they respond to. (A) LuxR solo responding to the intra-AHL signal of a noncognate LuxI synthase. (B) LuxR solo responding to an exogenous AHL signal. (C) LuxR solo responding to a non-AHL endogenous signal. (D) LuxR solo responding to an exogenous non-AHL exogenous signal. (E) LuxR solo acting in a signal-independent manner. (Created using Biorender.com.)

orphans or LuxR solos (12–14). The latter possess the same protein domain organization as canonical QS LuxRs (AHL-binding and DNA-binding); the DNA-binding domain is highly conserved, while the AHL-ligand-binding domain can present a few variations, possibly resulting in a response to non-AHL signals (13). Hence, owing to advances in machine

learning and experimentally determined structures, it is possible to predict structural and, thereby, functional similarities between related LuxR homologs (5).

LuxR solos are found in AHL-producing and non-AHL-producing proteobacteria which live in diverse environments, indicating that they have evolved throughout the proteobacterial phylum and likely play a major role in cell-cell signaling (30, 31). The few LuxR solos studied so far highlight that they can be involved in (i) sensing the host environment, (ii) sensing and responding to QS signals produced by competing or cooperating bacteria (e.g., AHLs), (iii) regulating self-target genes in a ligand-independent way or binding to yet-unknown molecules, and (iv) informing bacteria of their whereabouts, being not necessarily involved in QS (31, 32). Why are proteins so similar in their binding domain and tertiary structure so polyhedric? Do the surrounding environment and the ecological function put a selective pressure on the LuxR solos type and signal binding? A summary of the variability and signal versatility of the LuxR solo proteins studied and characterized so far is presented (Table 1). Below, we describe the 5 classes of LuxR solos that we have subdivided and depicted in Fig. 1.

## LuxR SOLOS RESPONDING TO ENDOGENOUS AHL SIGNALS (CLASS A)

Some LuxR solos respond to AHLs endogenously produced via the presence of a canonical LuxI/R QS system(s) in the same genome. This can extend the set of gene targets regulated by QS and result in the QS circuit being more adaptable by responding to other environmental cues. LuxR solos of this class include ExpR of *Sinorhizobium meliloti*, QscR of *Pseudomonas aeruginosa*, and BisR of *Rhizobium leguminosarum* bv. *viciae*, which are harbored in bacterial species that possess multiple canonical AHL QS systems. They all share a low homology with the resident QS LuxR proteins, suggesting that they have been acquired independently and are not a specific feature of one single taxa or closely related taxa.

QscR from *P. aeruginosa* is well characterized, having a relaxed AHL ligand-binding specificity (33, 34). In response to different AHL signals, QscR regulates different set of target genes, which can be distinct from those regulated by the two canonical QS systems (LasI/R and RhlI/R) of *P. aeruginosa*. One of its possible roles is the timing of the AHL response and virulence factor production, by regulating the adjacent phenazine biosynthetic operon (35). QscR has the highest specificity for 3-oxo-C10-AHL; however, LasI synthesizes 3-oxo-C12-AHL, suggesting that QscR responds and senses to a wide range of exogenous AHLs produced by neighboring bacteria.

Similarly, also, ExpR from *S. meliloti* is able to respond to a wide range of AHLs produced by the endogenous SinI/R QS system. ExpR is soluble and stable in the absence of AHLs and is versatile acting either as an activator or repressor depending on the presence or absence of AHLs. ExpR regulates a large set of target genes important for plant-growth promotion traits such as succinoglycan production, nitrogen fixation, motility, and metal transport (13, 36–39).

BisR from *R. leguminosarum* bv. *viciae* is well integrated with the three canonical QS systems resident in its genome (CinI/R, RaiI/R, and TraI/R), which produce several different AHLs (40). BisR is located on the symbiotic plasmid pRL1J1 and it is involved in growth inhibition in the presence of endogenously produced 3OH-C14:1-AHL. BisR negatively regulates the *cinI/R* QS system; CinI synthesizes $3O\text{-}C_{14}$-HSL, which BisR responds to in order to positively regulate *traR*, which induces conjugal transfer (41). BisR therefore ensures the induction of conjugal transfer only when a quorum of $3O\text{-}C_{14}$-AHL-producing bacteria is established. Phylogenetically, the BisR solo forms a monophyletic group specific to rhizobia and *Agrobacterium* spp., likely allowing cross talk between these species that share the same environmental niche (42).

Two other LuxR solos, called VjbR and BlxR from *Brucella melitensis*, also respond to endogenously produced AHL molecules. Interestingly, *B. melitensis* produces C12-AHL even though it does not possess a *luxI* homologue in its genome. Possibly, the production of the AHL signal could be due to the presence in its genome of an *hdtS* homolog

**TABLE 1** Functionally and molecularly characterized LuxR solos[a]

| LuxR solo protein | Bacterial species | Genomic context of LuxR solo (5′–3′) | Ligand type/binding molecule(s) | Type of signaling | LuxR class | Reference |
|---|---|---|---|---|---|---|
| NesR | *Sinorhizobium meliloti* | *pip*, proline iminopeptidase/*tra*, peptide transporter—*nesR*—*pip*, proline iminopeptidase/*hp* | Plant signal molecule | Interkingdom signaling with plants | D | 69 |
| OryR | *Xanthomonas oryzae* | *pip*, proline iminopeptidase/*tra*, peptide transporter—*oryR*—*pip*, proline iminopeptidase/*hp* | Plant signal molecule | Interkingdom signaling with plants | D | 66, 67 |
| PipR | *Pseudomonas* sp. GM79 | ala amino-peptidase *aapA*; *pipR*; pro-iminopeptidase, *pipA* | Ethanolamine derivative (HEHEAA) | Interkingdom signaling with plants | D | 70, 72, 73 |
| PsaR2 | *Pseudomonas syringae* pv. *actinidiae* | *pip*, proline iminopeptidase—*psaR2*—peptide/nickel transport system permease protein | Plant signal molecule | Interkingdom signaling with plants | D | 68 |
| PsoR | *Pseudomonas protegens* CHA0 | *pip*, proline iminopeptidase/*tra*, peptide transporter—*psoR*—*pip*, proline iminopeptidase/*hp* | Plant signal molecule from root exudates, i.e., saponarin and 2-benoxazolinone (BOA) | Interkingdom signaling with plants | D | 83 |
| PsrR | *Kosakonia* sp. | Glutathione trs.; *psrR*; *pip*, prolineiminopeptidase | Plant signal molecule | Interkingdom signaling with plants | D | 54 |
| XagR | *Xanthomonas axonopodis* | *pelB*, periplasmic pectate lyase; *xagR*; *pip*, prolineiminopeptidase | Plant signal molecule | Interkingdom signaling with plants | D | 84 |
| XccR | *Xanthomonas campestris* pv. *campestris* | *pip*, proline iminopeptidase/*tra*, peptide transporter—*xccR*—*pip,* proline iminopeptidase/*hp* | Plant signal molecule | Interkingdom signaling with plants | D | 71 |
| XocR | *Xanthomonas oryzae* | *pip*, proline iminopeptidase/*tra*, peptide transporter—*xocR*—*pip*, proline iminopeptidase/*hp* | Plant signal molecule | Interkingdom signaling with plants | D | 85 |
| MmsR | *Methylomonas* sp. strain LW13 | NAD(P)H-dependent FMN reductase; *mmsR*; *hp*, hypothetical protein | 3-OH-$C_{10}$-HSL; 3-OH-$C_{8}$-HSL; 3-OH-$C_{12}$-HSL | Interspecies signaling | B | 56 |
| BisR | *Rhizobium leguminosarum* | *trbl*, conjugal transfer protein Trbl—*bisR*—*traR*, conjugal transfer regulator TraR | Exogenous 3O-$C_{14}$-HSL | Interspecies signaling | B | 41 |
| LoxR | *Kosakonia* sp. | *yec*, ABC-transporter; *loxR*; hypothetical protein; *uvrY* | $C_{6}$-HSL and $OC_{12}$-HSL | Interspecies signaling | B | 54 |
| LuxR2 | *Pandoraea* sp. | *lysR*, LysR transcriptional regulator; *luxR2*; RND transporter | Exogenous HSLs | Interspecies signaling | B | 86 |
| LuxR3 | *Pandoraea* sp. | Membrane protein; *luxR3*; cytochrome *c* oxidase subunits I | Exogenous HSLs | Interspecies signaling | B | 86 |
| PpoR | *Pseudomonas putida* | *fprB*, flavodoxin reductase—*ppoR*- *rlmG*, 16s RNA, methylase RsmC | 3-oxo-$C_{6}$-AHL | Interspecies signaling | B | 55 |
| PsaR1 | *Pseudomonas syringae* pv. *actinidiae* | *hp*, hypothetical protein—*psaR1*—enolase-phospate E1 | Exogenous HSLs | Interspecies signaling | B | 68, 87 |
| PsaR3 | *Pseudomonas syringae* pv. *actinidiae* | *hp*, hypothetical protein—*psaR3*—anthranilate/para-aminobenzoate synthase component II | Exogenous HSLs | Inter-species signaling | B | 68, 87 |
| SmoR | *Stenotrophomonas maltophilia* | *hchA*; *smoR*; *hp*, hypothetical protein | Exogenous HSLs (oxo-$C_{8}$-HSL) | Interspecies signaling | B | 88 |
| VjbR | *Brucella melitensis* | *hp*, hypothetical protein—*vjbR*—*tetR*, TetR family transcriptional regulator | $C_{12}$-HSL | Interspecies signaling | B | 89 |

**TABLE 1** (Continued)

| LuxR solo protein | Bacterial species | Genomic context of LuxR solo (5′–3′) | Ligand type/binding molecule(s) | Type of signaling | LuxR class | Reference |
|---|---|---|---|---|---|---|
| SdiA | *Escherichia coli* and *Salmonella enterica* | *yec*, ABC-transporter—*sdiA*; *sirA*, invasion response regulator | 1-octanoyl-*rac*-glycerol; indole; 3-oxo-$C_8$-HSL; 3-oxo-$C_6$-HSL; 3-oxo-$C_{10}$-HSL; C6-HSL; $C_8$-HSL | Interspecies signaling | B | 46, 90–92 |
| QscR | *Pseudomonas aeruginosa* | Desaturase—*qscR*—*phzA*, phenazine biosynthesis protein A | 3-oxo-$C_{12}$-HSL and other signals (i.e., antibiotics); more sensitive to $3O$-$C_{10}$-HSL | Intra- and interspecies signaling | A | 33, 35 |
| NurR | *Sinorhizobium meliloti* | DNA-binding CsgD family transcriptional regulator —*nurR*—ATP-binding protein | Common metabolites (HSL-independent solo) | Intra- and interspecies signaling | A | 93, 94 |
| ExpR | *Sinorhizobium meliloti* | *chvA*, glucan exporter ATP-binding protein—*expR*—*pyc*, pyruvate carboxylase | $C_{14}$-HSL; 3-oxo-$C_{14}$-HSL; $C_{16:1}$-HSL; 3-oxo-$C_{16}$-HSL; $C_{18}$-HSL | Intraspecies signaling | C | 37–39 |
| PauR | *Photorhabdus asymbiotica* | *pauR*; pcfABCDEF operon | Dialkylresorcinols (DARs) | Intraspecies signaling | C | 57, 58 |
| PluR | *Photorhabdus luminescens* | *ppys*; *pluR*; pcfABCDEF operon | Photopyrones (PPYs) | Intraspecies signaling | C | 57, 58 |
| AnoR | *Achromobacter xylosoxidans* SD115 | *gbpR*; *anoR*; *xthA* | Not yet determined | Not yet determined | E | 95 |
| AvhR | *Agrobacterium vitis* | Calcium binding protein—*avhR*—*nrt*, nucleotide binding ABC transporter | Not yet determined | Not yet determined | E | 96 |
| AviR | *Agrobacterium vitis* | *chvA*, glucan exporter ATP-binding protein—*aviR*—*pyc*, pyruvate carboxylase | Not yet determined | Not yet determined | E | 97 |
| BlxR | *Brucella melitensis* | *ms*, methionine synthase—*blxR*—*at*, amino transferase | Not yet determined | Not yet determined | E | 39 |
| CarR | *Serratia marcescens* | Gene X, any gene product—*carR*—*carA*, carbapenem biosynthesis protein | Antibiotics (HSL- independent solo) | not yet determined | E | 74, 98 |
| CepR2 | *Burkholderia cenocepacia* | *araC*, AraC family transcriptional regulator —*cepR2*—gene X, any gene product | Common metabolites (HSL-independent solo) | Not yet determined | E | 75, 99 |
| MalR | *Burkholderia thailandensis* | *mal* gene cluster, involved in the production of the cytotoxic polyketide malleilactone | Antibiotics such as trimethoprim; it functions also independently of any external or endogenous signal | Not yet determined | E | 76 |
| RpaR | *Methylobacterium sp.* SD274 | DUF488 domain-containing protein; *rpaR*; *emaA* | Not yet determined | Not yet determined | E | 95 |
| VisR | *Sinorhizobium meliloti* | Pyruvate carboxylase—*visR*—transposase | Not yet determined | Not yet determined | E | 100 |

[a]trs, transporter; RND, resistance-nodulation-cell division family of transporter; HSL, acyl-homoserine lactone.

gene, which could be responsible for synthesizing AHLs or via the presence of a novel AHL synthetase that remains to be identified (39, 43–45).

In summary, LuxR solos responding to endogenous AHLs render QS more adaptable and can possibly respond to AHLs produced by neighboring bacteria being involved also in interspecies signaling. They could also affect AHL levels by creating a threshold needed for the activation of the native QS systems. Alternatively, they could possibly ensure that there is no cellular AHL leak-out, resulting in an over-response.

## LuxR SOLOS RESPONDING TO EXOGENOUS AHL SIGNALS (CLASS B)

The number of predicted LuxR solos in non-AHL-producing proteobacteria is astonishingly high; thus, many probably eavesdrop on AHLs present within the microbial community. While the QS AHL-binding LuxRs communicate with the next of kin (two-way

communication), the AHL-binding LuxR solos harbored by non-AHL-producing bacteria may provide cues on the bacterial population structure and density by activating or repressing target genes accordingly (one-way communication or signal interception) (46). Only a few LuxR solos able to bind and respond to exogenous AHLs have thus far been characterized. The most studied is SdiA, which is conserved across many enteric bacteria such as, *Salmonella*, *Escherichia*, *Klebsiella*, *Shigella*, and *Enterobacter* (46–48). SdiA, compared to other QS LuxRs, is unique in being stable *in vitro* in its apo-form (i.e., unbound to AHLs) and is rather promiscuous and able to bind a broad range of AHLs (49, 50). Several SdiA gene targets have been determined, most prominently in *E. coli*, where in the stomach environment of cattle, it activates the *gad* operon in response to AHLs produced by commensal or pathogenic bacteria (e.g., *P. aeruginosa*, *Aeromonas hydrophila*), which contributes to *E. coli* becoming acid tolerant (51–53). SdiA gene expression is then altered in the intestine since it is an alkaline environment mostly devoid of AHLs, and consequently, certain negatively regulated SdiA/AHL loci become de-repressed, allowing attachment of *E. coli* and the formation of lesions, which facilitates its shedding in the environment (51, 52). Recently, the SdiA ortholog LoxR harbored by the plant-beneficial endophyte *Kosakonia* spp. has been shown to promiscuously respond to exogenously produced AHLs; hence, SdiA also plays a role in the colonization of the plant environment (54).

Another example is the *P. putida* PpoR solo, which binds to exogenous 3-oxo-C6-AHL; it is conserved among *P. putida* species, being present in AHL-producing as well as non-AHL-producing strains. Transcriptome studies indicated that it is involved in inorganic ion utilization and response to oxidative stresses (55). Similarly, MmsR, which is encoded by several species of methane-oxidizing bacteria belonging to the genus *Methylomonas*, responds to multiple exogenous acyl-AHL signals, including 3-OH-C10-AHL. The latter AHL is produced by the QS system of *Methylobacter tundripaludum*, which is another methane oxidizer bacterium sharing the same anaerobic ecosystem. Thus, via MmsR, two bacterial species interact and cooperate or compete for resources in the same environment (56).

In summary, this type of LuxR solo could serve as sensors for the entire microbial community, having important roles in shaping and/or maintaining the functional structure of the microbiome. On the other hand, they can be used to sense the presence of predators and/or competitors and react accordingly.

## LuxR SOLOS RESPONDING TO NON-AHL ENDOGENOUS SIGNALS (CLASS C)

Some LuxR solos bind to non-AHL compounds which are endogenously produced, hence creating regulatory circuits similar to the canonical AHL QS systems but using different signal-generator synthases and cognate ligand compounds. For example, the two LuxR solos PluR and PauR from *Photorhabdus luminescens*, an insect pathogen, and *Photorhabdus asymbiotica*, a human enteric and insect pathogen, are part of a novel cell-to-cell signaling system which respond to the endogenously produced photopyrones (PPY) and dialkylresorcinols (DARs), respectively. These signal molecules activate via PluR and PauR the expression of the nearby operon, leading to cell clumping and contributing to the virulence of *Photorhabdus* species (57, 58). Interestingly, these two signals are bifunctional; PPYs can act as insect toxins at high concentrations, and DARs can act as antibiotics (58). The PPY cognate signal synthase of PluR is synthesized by the adjacent gene *ppyS*, having therefore, a similar organization of the canonical *luxI/R* system (59). However, in both PluR/PpyS and PauR/DarABC the signal synthase genes are not under the control of the LuxR homologue, and the systems have no autoinduction, which is one of the QS hallmarks. PluR and PauR both harbor four substitutions at similar positions in the conserved WYDPWG-6-motif of AHL sensors, displaying a TYDQCS-motif and a TYDQYI-motif, respectively, which most likely allows these solos to respond no longer to AHLs but to PPY and DAR (58).

In summary, the number of LuxR solos that bind and respond to non-AHL endogenous compounds is likely to increase in view that many *luxR* solo genes are flanked by uncharacterized genes coding for synthases or biosynthetic operons which probably

synthesize unknown small-molecular-weight compounds that will likely act as signals for the adjacent LuxR solo (see below).

## LuxR SOLOS RESPONDING TO NON-AHL EXOGENOUS SIGNALS (CLASS D)

Some LuxR-family regulators have evolved to bind to non-AHL signal molecules, in some cases to signals which are similar to AHLs, for example, (i) RpaR from *Rhodopseudomonas palustris*, which binds and responds to *p*-coumaroyl-homoserine lactone derived from the exogenously provided plant metabolite *p*-coumarate (60, 61) and (ii) BraR from *Bradyrhizobium*, which responds to cinnamoyl-homoserine lactone (62). Signal specificity can be altered by specific changes in some key residues in the ligand-binding domain of LuxR receptors (63). A subclass of LuxR solos found only in plant-associated bacteria (PAB), both beneficial and pathogenic, has evolved to respond to plant signals (30, 64, 65). These lack some conservation in the 6 key residues of the AHL-binding domain (66, 67), more precisely, W57 and Y61 (with respect to TraR), which are substituted by M and W, respectively. This subfamily of PAB LuxR solos is involved in the colonization of either a plant pathogen or beneficial bacterium (64). Members include XccR of *Xanthomonas campestris*, OryR of *Xanthomonas oryzae*, PsoR of *Pseudomonas fluorescens*, XagR of *Xanthomonas axonopodis*, NesR in *Sinorhizobium meliloti*, PipR of *Pseudomonas* sp. strain GM79, PsrR of *Kosakonia* spp., and PsaR2 of *Pseudomonas syringae* pv. *actinidiae* (13, 54, 66–71). A universal feature of this interkingdom plant-bacterium circuit is that a proline iminopeptidase (*pip*) gene(s) is regulated and located next to the PAB LuxR solo (70). The *pip* genes have been implicated in virulence factors, but their mode of action remains unknown (71). Phylogenetically, they are highly related and cluster together regardless of the taxonomy, further confirming an ecological role (54). PipR, from the *Populus deltoides* root endophyte *Pseudomonas* sp. GM79, activates the downstream *pip* gene in response to an ethanolamine derivative (HEHEAA) which forms spontaneously from ethanolamine and serves as an intermediate in plant cell membrane biogenesis and plant hormones (72, 73). Recent studies have resolved the crystal structure of an ABC transporter which is required for import of the HEHEAA compound into bacterial cells (73); however, a close homolog of the transporter protein in another bacterial species cannot bind HEHEAA, implying that there are other effector compound(s) for this widespread PAB signaling system.

In summary, PAB LuxR solos as well as SdiA-like solos represent two widespread and conserved LuxR solo subgroups which share the same genetic arrangement among different taxonomically distant bacterial genera. These two subgroups have an ecological-related function since they are present among taxonomically different bacteria that, however, live in and colonize similar habitats. In view of the high number of distinct types of LuxR solo sequences and clades (see below), it is highly likely that many other functional and ecological LuxR solo subgroups exist in bacteria.

## LuxR SOLOS ACTING IN A LIGAND-INDEPENDENT MANNER (CLASS E)

Some LuxR solos retain all of the 6 key conserved residues in their AHL-binding domain; however, they do not bind and respond to AHLs and regulate gene expression in a ligand-independent manner. MalR from *Burkholderia thailandensis*, CarR from *Serratia marcescens*, and CepR2 from *Burkholderia cenocepacia* are three examples (74–76). Their mechanism of regulation is not yet well understood; the activation of the target genes depends on sufficient transcription of the *luxR* genes (76).

This further expands the number of different regulatory scenarios used by LuxR solos; in this case the regulation of the LuxR target genes is likely dependent on the environmental response of other additional regulators, which in turn are responsible for the transcriptional and/or posttranscriptional regulation of the *luxR* solo genes.

## *IN SILICO* ANALYSIS OF THE DISTRIBUTION, FUNCTIONAL GROUPING, AND ECOLOGICAL CLASSIFICATION OF THE LuxR SOLO HOMOLOGS AMONG THE PROTEOBACTERIA

A few bioinformatic and metagenomic studies have shown that some proteobacterial genomes harbor an excess of QS *luxR* family transcriptional regulators compared to QS *luxI* homologs, implying that the *luxR* solos are more abundant and predominant

than complete *luxI/R* QS systems (15, 16, 42). Multiple LuxR solos can also be present in the same genome, showing different levels of relatedness and suggesting different binding properties, biological roles, origins, and evolution (16). Recently, an analysis of approximately 600 genomes of environmental fluorescent pseudomonads revealed that LuxR solos are predominant, with over 50% harboring at least one *luxR* solo gene (77). Primary structure and adjacent loci allowed the subdivision of the majority of these pseudomonad LuxR solos into several subgroups. Analysis of the AHL-binding domain and fold structural prediction indicated that most subgroups likely bind to currently unknown signal molecules and a few to AHLs (77).

In order to map LuxR solo distribution among all proteobacteria sequenced thus far, we performed a bioinformatic analysis and present it here. A collection of 31,034 proteobacterial genomes belonging to 1,042 bacterial genera was searched for the presence of LuxR solos containing its signature Pfam domain (PF03472 autoind_bind domain) and characterized by the absence, in the surrounding 8 genes (4 genes upstream and 4 genes downstream of LuxR), of the AHL-synthase domain (PF00765), defining the LuxI proteins. In total, 26,577 *luxR* solo hits were identified in 16,683 genomes; out of 31,034 genomes analyzed, $\sim$46.2% ([31,034 − 16,683]/31,034) do not harbor any *luxR* solos, $\sim$39.4% (12,235/31,034) encode one *luxR* solo gene, and $\sim$14.3% (4,448/31,034) encode more than one *luxR* solo. In order to reduce the complexity and remove very closely related protein sequences, a clustering was performed that shared 80% identity over 80% of coverage, resulting in 4,276 clusters. Bacterial genera such as *Rhizobium*, *Pseudomonas*, *Burkholderia*, and *Agrobacterium* carry the highest number of different LuxR solo clusters (Fig. 2A). On the other hand, proteobacterial genera such as *Escherichia*, *Erwinia*, and *Klebsiella* harbor only a few types of LuxR solos, suggesting a high level of conservation.

We then further stringently grouped the LuxR clusters by the genetic context (specifically, of the pfam domains) surrounding the cluster members (Fig. 2B). We classified 341 LuxRs that fit the criteria for grouping (Fig. 2B), which represent the most prevalent type of LuxR solos among proteobacteria, but not unique. The results showed that 274 LuxR clusters separated into 8 subgroups, while 67 could not be classified. The LuxR solos do not always group according to the taxonomy, since several branches of the tree are formed by LuxR solo orthologs belonging to different bacterial genera. Moreover, LuxR solos often but not always group according to their flanking genetic context; it is possible that highly similar LuxR solos in the primary structure are acquired by different bacteria but not necessarily involved in the regulation of the same target genes or participating in the same biological processes. The majority of them belong to the subgroup 2; all the solo hits classified in this group share a hallmark that is the presence in their vicinity of the gene *mfs1* coding for a major facilitator superfamily transporter 1 protein, and the majority of them belong to aquatic environments.

We noticed that the subgroups 0 and 1 are formed by not-closely related species which, however, showed conservation of the *luxR* solos flanking genetic regions. In particular, the subgroup 0 is formed by PAB ortholog solos that respond to plant signals (see above), which have been detected in several plant-associated bacterial genera such as *Pseudomonas*, *Rhizobium*, *Agrobacterium*, *Shinella*, *Aureimonas*, *Enterobacter*, *Rhodobacter*, *Dickeya*, *Xanthomonas*, and *Kosakonia*. All these genera have very conserved primary sequences and flanking gene arrangements. Similarly, the subgroup 1 consists of SdiA ortholog solos belonging to members of the *Enterobacteriaceae* family such as *Escherichia*, *Shigella*, *Salmonella*, *Enterobacter*, *Citrobacter*, *Cedecea*, *Buttiauxella*, *Franconibacter*, *Cronobacter*, and *Yokenella*, which share the same flanking gene arrangement. Interestingly, we also observed for these two subgroups a correlation with the ecological role, as all the solos classified in the subgroup 0 have been detected in bacteria isolated from plant or terrestrial environments, while all the solo hits classified in the subgroup 1 have been detected in human- or terrestrial-isolated bacteria (Fig. 2B).

On the other hand, the subgroups 3, 5, 6, and 7 are formed by solo LuxRs carried by closely related bacteria showing conservation of the LuxR solo flanking genes, while not necessarily the same niche-specific distribution. In particular, the solo LuxRs clustered in the subgroup 3, 5, or 6 belong to the *Rhizobiaceae* family (*Rhizobium*, *Mesorhizobium*,

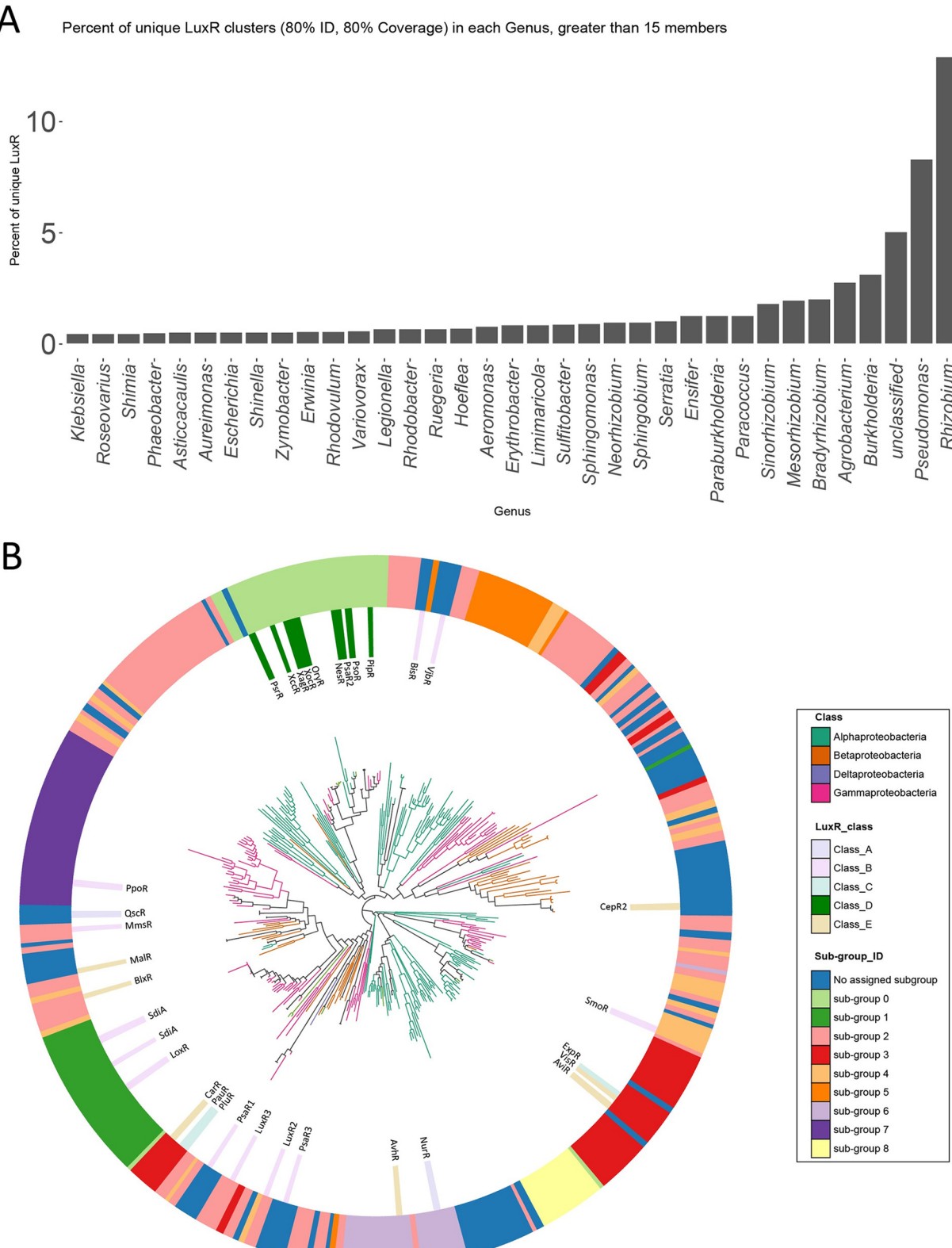

**FIG 2** Taxonomic distribution and classification of LuxR solos in proteobacterial genomes. Genomes of proteobacteria from the Integrated Microbial Genomes (IMG) database ($n$ = 31,034, (78)) were searched for the presence of *luxR* solo genes (those that encode for proteins with domain PF03472). Four genes up- and downstream of each gene were checked for the absence of *luxI* (PF00765). The proteins were clustered with CD-HIT (79) at 80% identity and 80% coverage of both query and subject, resulting in 4,276 LuxR clusters that we considered distinct. (A) Each genus ($x$ axis) was counted for the number of unique LuxR clusters, and those with 15 or more unique members were

*Sinorhizobium, Bradyrhizobium, Ensifer,* and *Shinella*) and they represent a specific signature of this group of bacteria. Similarly, the subgroup 7 consists of LuxR solos belonging to the *Pseudomonas* genus only, presenting a conserved flanking gene arrangement but different levels of relatedness. Also in this case, these solos are well conserved and mostly harbored on genomes of *Pseudomonas* which have been isolated from many different environments spanning from aquatic to plant locations (Fig. 2B), suggesting a role for these proteins in adaptation of environmental pseudomonads to diverse habitats.

The bioinformatic survey presented here shows exclusively the most prevalent and representative LuxR solo subgroups from proteobacteria. By relaxing the search and grouping criteria, many more subgroups will be generated which are less abundant and shared among bacteria but which have some variability in the adjacent genomic context and still retain significant relatedness.

In addition, to further investigate whether LuxR solo homologs can also cluster according to the type of signal molecules they bind to, we added to this bioinformatic analysis the solo LuxR proteins that have already been molecularly characterized and studied as described in Table 1 and presented in Fig. 2B. Interestingly, we noticed a correlation between the LuxR solos belonging to the class D and known to respond to plant-derived molecules and the LuxR solos belonging to the subgroup 0, characterized by the presence of a proline iminopeptidase (*pip*) gene(s) located next to each PAB LuxR solo. For all the other LuxR solos already studied, we did not observe any direct correlation between the type of ligand they respond to and their genetic synteny or primary structure.

## FUTURE PROSPECTS

The present review highlighted that LuxR solos in proteobacteria are exceptionally abundant that and many diverse subgroups exist, having different evolutionary origins, functions, ligand binding properties, and ecological contexts. The solos suggest that LuxR-type QS may have evolved in many forms over evolutionary time, yet we know little about the present variety and evolutionary history. Some types of solo LuxRs are likely to be specialized in relation to the taxonomy, others to the regulatory mechanisms they might participate in, and others to the specific niche they occupy. Future work needs to validate their role in intraspecies, interspecies, and interkingdom signaling in bacterial communities as well as the chemical signals they respond to in order to unravel the function of this major regulatory family in cell-cell interactions of the microbiome establishment and maintenance. Moreover, future studies need to investigate, via structure-based search, for *luxI* and *luxR* genes in host genomes (e.g., plants, insects, fungi, mammals) to test whether the hosts acquired these genes to improve interactions with their microbes. This analysis may help in the identification of the source of the signals received by some LuxR solos.

## ACKNOWLEDGMENTS

A.L. is generously supported by the Israeli Science Foundation (grant no. 1535/20 and 3300/20), the Alon Fellowship of the Israeli council of higher education, The Hebrew University–University of Illinois Urbana-Champaign seed grant, the Israeli Ministry of Agriculture (grant 12-12-0002), and ICA in Israel. A.M.G. is supported by the Kaete Klausner scholarship. C.B. and V.V. are supported by ICGEB.

**FIG 2** Legend (Continued)

graphed. (B) The surrounding genes in the *luxR* gene clusters were retrieved, as well as the pfam domains in the proteins encoded by these genes. Clusters with more than 50 members were further analyzed by counting the frequencies of appearances of their pfams. Pfams with less than 5% appearance in the cluster surroundings were dropped from the analysis. The frequencies of appearance were then dimensionally reduced using UMAP (n_components = 3). The dimensionally reduced data were then clustered using DBSCAN (EPS = 0.5, min_samples = 10), resulting in 8 subgroups of LuxR (and those that were not clustered, "subgroup" −1). A tree of the LuxR cluster representatives was made by aligning their protein sequences using Clustal Omega with standard settings (80) and then using FastTree with the alignment as input with standard settings (81). The tree is annotated with class (see legend on top on the right), LuxR classes as presented in Table 1 (see legend middle-right), and with the subgroup ID (internal ring; see legend on the right). The names of the best-characterized LuxR solos so far are indicated in the tree and colored according to the LuxR classes they belong to. The tree was visualized and annotated using EMPress (82).

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
