## [Reviewer comments · mSystems]

Cell-cell signaling proteobacterial LuxR solos: a treasure trove of subgroups having different origins, ligands and ecological roles

Cristina Bez, Alex Geller, asaf levy, and Vittorio Venturi

Corresponding Author(s): Vittorio Venturi, International Centre for Genetic Engineering and Biotechnology

Review Timeline:

Submission Date:	October 21, 2022
Editorial Decision:	November 10, 2022
Revision Received:	December 20, 2022
Accepted:	December 22, 2022

Editor: Jack Gilbert

Reviewer(s): Disclosure of reviewer identity is with reference to reviewer comments included in decision letter(s). The following individuals involved in review of your submission have agreed to reveal their identity: Leo Eberl (Reviewer #1); Michael A Savka (Reviewer #3)

Transaction Report:

DOI: <https://doi.org/10.1128/msystems.01039-22>

Dr. Vittorio Venturi
International Centre for Genetic Engineering and Biotechnology
Bacteriology Lab
Padriciano, 99
Padriciano 99
Trieste, Trieste 34149
Italy

Re: mSystems01039-22 (Cell-cell signaling proteobacterial LuxR solos are prevalent and drifting away from AHLs)

Dear Dr. Venturi:

Reviewer comments are found at the end of this letter.

Your minireview is likely to be accepted once the indicated changes are made.

Author Bios: If you would like a brief biographical sketch of each author (limit, 150 words) to be published at the end of your article, please submit text and photos with your modified manuscript. For complete guidelines on submission requirements, please see the journal Submission and Review Process requirements at <https://journals.asm.org/journal/mSystems/submission-review-process>. **Submissions of a paper that does not conform to mSystems guidelines will delay acceptance of your manuscript.**

Figures [**Editor: insert figure numbers here**] in your manuscript are good candidates for graphical enhancement. We now offer our authors the services of ASM's contracted artist, Patrick Lane of ScEYence Studios. This art enhancement service is free of charge to authors of minireviews and full-length reviews, and turnaround time is fast. Please contact Patrick on receiving this letter. Complete contact information for Patrick and further instructions are posted at <https://journals.asm.org/pb-assets/pdf-text-excel-files/graphical-enhancement-support.pdf>

Please return your modified manuscript within 60 days; if you cannot complete the modification within this time period, please contact me. If you decide that you do not want to modify the manuscript and wish to submit it to another journal, please notify me of your decision immediately so that the manuscript can be formally withdrawn.

To submit the modified manuscript, log onto the eJP submission site at <https://msystems.msubmit.net/cgi-bin/main.plex>. If you cannot remember your password, click the "Can't remember your password?" link and follow the instructions on the screen. Go to Author Tasks and click the appropriate manuscript title to begin the resubmission process. The information you entered when you first submitted the paper will be displayed. Please update the information as necessary. Provide (1) point-by-point responses to the issues raised by the reviewers as file type "Response to Reviewers," not in your cover letter, and (2) a PDF file that indicates the changes from the original submission (by highlighting or underlining the changes) as file type "Marked Up Manuscript - For Review Only."

To submit your modified manuscript, log onto the eJP submission site at <https://msystems.msubmit.net/cgi-bin/main.plex>. If you cannot remember your password, click the "Can't remember your password?" link and follow the instructions on the screen. Go to Author Tasks and click the appropriate manuscript title to begin the resubmission process (ONLY the corresponding author will have access to the full record for resubmission). The information that you entered when you first submitted the paper will be displayed. Please update the information as necessary and do the following:

- 1) Provide point-by-point responses to the issues raised by the reviewers in a file designated as "Response to Reviewers" (NOT the cover letter).
- 2) Upload ALL of your source files (not PDF and not just the files requiring modification) and make sure that all elements meet the technical requirements for production.
- 3) Do not provide a highlighted or tracked-changes copy of the paper in the main manuscript upload. This should be a clean copy instead. You may provide the compare copy separately by uploading it as a "Marked-Up Manuscript" file.
- 4) Make sure that the figure legends are included in the main manuscript file (not uploaded separately).

ASM policy requires that data be available to the public upon online posting of the article, so please verify all links to sequence records, if present, and make sure that each number retrieves the full record of the data. If a new accession number is not linked or a link is broken, provide production staff with the correct URL for the record. If the accession numbers for new data are not publicly accessible before the expected online posting of the article, publication of your article may be delayed; please contact

the ASM production staff immediately with the expected release date.

If you would like to submit an image for consideration as the Featured Image for an issue, please contact mSystems staff.

Sincerely,

Jack Gilbert
Editor, mSystems

Journals Department
Reviewer comments:

Reviewer #1 (Comments for the Author):

This review summarizes the current knowledge on LuxR solos, i.e. LuxR-type regulators that are disconnected from their cognate signal synthases. While there are several reviews on this topic this article, based on some new, interesting bioinformatic analyses, provides a refreshing look at the topic. I am convinced that the reported classification/grouping of LuxR solos will be of interest in the wider field of bacterial signaling. I have only minor points for further improvement of this well-done manuscript.

Fig. 2B: I think this figure could be greatly improved if the position of the better characterized LuxR solos shown in Table 1 would be indicated in this figure (at least some of them).

Would the members of the 5 classes of LuxR solos cluster in this tree? And how do the 5 classes of LuxR solos relate to the 8 subgroups that were defined on the basis of gene synteny? I think that this should be discussed in better detail.

- l. 35: please explain what RI, RMI, RLI and RXMI topologies are
- l. 53: delete "the totality of"
- l. 89: aa numbering is according to LuxR ? please clarify
- l. 149: It would be easy to look for ainS and hdtS homologs in the genomes of *B. melitensis*. Are homologs present?
- l. 193-: I do not understand why RpaR and BraR are considered LuxR solos as the cognate synthase genes are next to the genes encoding the regulators. The LuxI-type enzymes may need precursors for signal biosynthesis from the environment but their cognate receptors are not solos according to the definition given in the introduction.
- l.278: more than one luxR solo
- l. 290: remove do

Reviewer #2 (Comments for the Author):

The article „Cell-cell signaling proteobacterial LuxR solos are prevalent and drifting away from AHLs" from Bez et al. is a review about a bacterial receptor family that is known to be involved in cell-cell communication and interkingdom signaling. The article is well written and the newest literature on the topic of LuxR solos is cited. I only have a few minor points that the authors should address:

- Lines 35-36.: RI, RMI, RLI topology. These abbreviations and the topology behind should be explained.
- Lines 148-151. It is mentioned that *B. melitensis* produces C12-AHL even though it does not possess a luxI homologue in its genome, which is interesting. However, the three papers cited (Ref. 45-47) are not examples for proteins involved in AHL synthesis independent from LuxI as cited. Protein AinS synthesizes AI-2, which is a furanosyl borate di-ester and not an AHL, and Bjal is an LuxI homologue. Please cite the correct literature.
- Line 229: Only *P. asymbiotica* is human and insect pathogenic, not *P. luminescens* (insect pathogen only). Please correct.
- Lines 235, 236: PluR/PpyS and PauR/DarABC are note completely similar to a classical LuxR/LuxI system, as both the signal synthase genes are not under control of the LuxR homologue and therefore the systems have no autoinduction. For clarity, this should be mentioned in the text.
- In addition to the two examples given for endogenous non-AHL signals, 3,5-dimethylpyrazin-2-one (DPO) was described as autoinducer in *Vibrio cholerae*. DPO is sensed at a certain threshold by the LuxR solo VqmA through its PAS4 signal domain, which induces the expression of vqmR to regulate the pathogenicity of *V. cholerae* (doi: 10.1038/nchembio.2336). I suggest

including this example into the review as to date PluR, PauR and VqmA are the only three examples known for endogenous non-AHL signals.

- L 274. The gene syntax luxR instead of the protein syntax LuxR should be used here.

- Table 1: The protein names in column 1 of the table shouldn't be in italics style. Also use the correct style (column 3) when gene or protein names are used (e.g. italics for gene names). Column 3: species names should all be in italics style.

- Fig. 2A: The genus names of the bacteria should be in italics style.

Reviewer #3 (Comments for the Author):

Excellent study of the known LuxR solos proteins. I enjoyed reading the manuscript, as it is well development and well written. I especially took interest to the categories of functional classes represented in Figure 1. This figure provides the current state of LuxR solos identified in the whole genomes of bacteria. Likewise, Table 1 provides a up-to-date identification of members within each of the functional classes. This provides the reader with a rich list of information to potential "jump-start" studies to identify and describe mechanisms the LuxR solo proteins participate in the development and succession of complex microbial communities.

The end of the title: "...and drifting away from AHLs" may be a bit narrow of a phrase to describe the outcomes of this study. Perhaps a broader phrase: "...prevalent: a treasure trove of subgroups, evolutionary origins, functions, ligands and ecological niches".

Line 323, the word "functionality" appears to be not correctly used in this context. Please re-phrase.

Line 329, the word "eons" should be replace with "over evolutionary time".

Line 331, the words "participate to" change to 'participate in".

Line 334, the words "function of a" replace with "function of this".

Thank you for your work and submission.

Response to Reviewers

Indicated below is a point-by-point response on how we have dealt the suggestions and comments of the two reviewers.

Reviewer #1 Fig. 2B: I think this figure could be greatly improved if the position of the better characterized LuxR solos shown in Table 1 would be indicated in this figure (at least some of them).

Authors' response; we thank the reviewer for raising this point that further improved the figure. We have now revised the Figure 2B adding the better characterized LuxR solos in the first ring of the tree colored according to the type of the 5 LuxR solo classes. We have now also better defined the 5 LuxR solo classes throughout the paper.

Reviewer #1 Would the members of the 5 classes of LuxR solos cluster in this tree? And how do the 5 classes of LuxR solos relate to the 8 subgroups that were defined on the basis of gene synteny? I think that this should be discussed in better detail.

Authors' response; we thank the reviewer for this suggestion and we have now improved the discussion of this point (l. 340-347)

Reviewer #1

l. 35: please explain what RI, RMI, RLI and RXMI topologies are

Authors' response; we thank the reviewer for raising this issue as we did not explain these topologies in the original submission. We now explain them in the revised version (l. 40-45)

Reviewer #1

l. 53: delete "the totality of"

l. 89: aa numbering is according to LuxR ? please clarify

Authors' response; we modified these points according to the reviewer suggestions.

Reviewer #1 l. 149: It would be easy to look for ainS and hdtS homologs in the genomes of B. melitensis. Are homologs present?

Authors' response; we agree and thank the reviewer for this suggestion: we have now performed a search in the genome of *B. melitensis* where we found a homolog (36% of aa homology) with hdtS gene. We have now revised the sentence accordingly (l. 53-55)

Reviewer #1 l. 193-: I do not understand why RpaR and BraR are considered LuxR solos as the cognate synthase genes are next to the genes encoding the regulators. The LuxI-type enzymes may need precursors for signal biosynthesis from the environment but their cognate receptors are not solos according to the definition given in the introduction.

Authors' response; we thank the reviewer for this comment. The sentence has now been rephrased (l.253)

Reviewer #1

l.278: more than one luxR solo

l. 290: remove do

Authors' response; We have dealt with these minor points.

Reviewer #2 Lines 35-36.: RI, RMI, RLI topology. These abbreviations and the topology behind should be explained.

Authors' response; we agree with the reviewer and we have now revised the sentence accordingly, as suggested also by reviewer 1. (l. 40-45)

Reviewer #2

Lines 148-151. It is mentioned that *B. melitensis* produces C12-AHL even though it does not possess a luxI homologue in its genome, which is interesting. However, the three papers cited (Ref. 45-47) are not examples for proteins involved in AHL synthesis independent from LuxI as cited. Protein AinS synthesizes AI-2, which is a furanosyl borate di-ester and not an AHL, and Bjal is an LuxI homologue. Please cite the correct literature.

Authors' response; we thank the reviewer for this observation. We have now revised the literature accordingly (l. 55).

Reviewer #2

Line 229: Only *P. asymbiotica* is human and insect pathogenic, not *P. luminescens* (insect pathogen only). Please correct.

Authors' response; thank you for pointing this out and we modified the revised version accordingly.

Reviewer #2

Lines 235, 236: PluR/PpyS and PauR/DarABC are not completely similar to a classical LuxR/LuxI system, as both the signal synthase genes are not under control of the LuxR homologue and therefore the systems have no autoinduction. For clarity, this should be mentioned in the text.

Authors' response; we thank the reviewer for raising this point. We have now clarified this very interesting aspect (l. 210-212)

Reviewer #2

In addition to the two examples given for endogenous non-AHL signals, 3,5-dimethylpyrazin-2-one (DPO) was described as autoinducer in *Vibrio cholerae*. DPO is sensed at a certain threshold by the LuxR solo VqmA through its PAS4 signal domain, which induces the expression of vqmR to regulate the pathogenicity of *V. cholerae* (doi: 10.1038/nchembio.2336). I suggest including this example into the review as to date PluR, PauR and VqmA are the only three examples known for endogenous non-AHL signals.

Authors' response; we thank the reviewer for this observation. However, in this review we defined as LuxR solos the LuxR protein regulators having both the Pfam domains (PF03472 autoind_bind domain and PF00196 DNA-binding helix-turn-helix) and characterized by the absence, in the surrounding 8 genes (4 genes upstream and 4 genes downstream of LuxR), of the AHL-synthase domain (PF00765). Hence LuxR solos are very close and evolved from QS LuxRs. For this reason we are not dealing with all the other type of proteins annotated as LuxR-like regulators which do not cluster in this sub-family.

Reviewer #2

L 274. The gene syntax luxR instead of the protein syntax LuxR should be used here.

Table 1: The protein names in column 1 of the table shouldn't be in italics style. Also use the correct stye (column 3) when gene or protein names are used (e.g. italics for gene names).

Column 3: species names should all be in italics style.

Authors' response; In the revised version, we have modified the Table 1 accordingly and corrected throughout the manuscript.

Reviewer #2

Fig. 2A: The genus names of the bacteria should be in italics style.

Authors' response; we have now revised the Figure 2A as suggested by the reviewer.

Reviewer #3

The end of the title: "...and drifting away from AHLs" many be a bit narrow of a phase to describe the outcomes of this study. Perhaps a broader phrase: "...prevalent: a treasure trove of subgroups, evolutionary origins, functions, ligands and ecological niches".

Authors' response; we thank the reviewer for this important comment and we have now modified the title following her/his suggestion.

Reviewer #3

Line 323, the word "functionality" appears to be not correctly used in this context. Please re-phase.

Authors' response; we have now revised the sentence accordingly.

Line 329, the word "eons" should be replace with "over evolutionary time".

Authors' response; we have now revised the sentence accordingly.

Line 331, the words "participate to" change to 'participate in".

Authors' response; we have now revised the sentence accordingly.

Line 334, the words "function of a" replace with "function of this".

Authors' response; we have now revised the sentence accordingly.

December 22, 2022

Dr. Vittorio Venturi
International Centre for Genetic Engineering and Biotechnology
Bacteriology Lab
Padriciano, 99
Padriciano 99
Trieste, Trieste 34149
Italy

Re: mSystems01039-22R1 (Cell-cell signaling proteobacterial LuxR solos: a treasure trove of subgroups having different origins, ligands and ecological roles)

Dear Dr. Venturi:

Your manuscript has been accepted, and I am forwarding it to the ASM Journals Department for publication. For your reference, ASM Journals' address is given below. Before it can be scheduled for publication, your manuscript will be checked by the mSystems production staff to make sure that all elements meet the technical requirements for publication. They will contact you if anything needs to be revised before copyediting and production can begin. Otherwise, you will be notified when your proofs are ready to be viewed.

Publication Fees:

If you would like to submit a potential Featured Image, please email a file and a short legend to mSystems@asmusa.org. Please note that we can only consider images that (i) the authors created or own and (ii) have not been previously published. By submitting, you agree that the image can be used under the same terms as the published article. File requirements: square dimensions (4" x 4"), 300 dpi resolution, RGB colorspace, TIF file format.

We recognize that the video files can become quite large, and so to avoid quality loss ASM suggests sending the video file via <https://www.wetransfer.com/>. When you have a final version of the video and the still ready to share, please send it to mSystems staff at mSystems@asmusa.org.

Sincerely,

Jack Gilbert
Editor, mSystems

Journals Department
E-mail: mSystems@asmusa.org